# Analytical Challenges in Diabetes Management: Towards Glycated Albumin Point-of-Care Detection

**DOI:** 10.3390/bios12090687

**Published:** 2022-08-26

**Authors:** Andrea Rescalli, Elena Maria Varoni, Francesco Cellesi, Pietro Cerveri

**Affiliations:** 1Department of Electronics, Information and Bioengineering, Politecnico di Milano, 20133 Milan, Italy; 2Department of Biomedical, Surgical and Dental Sciences, Università degli Studi di Milano, 20122 Milan, Italy; 3Department of Chemistry, Materials and Chemical Engineering “Giulio Natta”, Politecnico di Milano, 20133 Milan, Italy

**Keywords:** diabetes mellitus, glycemic control, glycated albumin, point-of-care testing, enzymes, antibodies, aptamers

## Abstract

Diabetes mellitus is a worldwide-spread chronic metabolic disease that occurs when the pancreas fails to produce enough insulin levels or when the body fails to effectively use the secreted pancreatic insulin, eventually resulting in hyperglycemia. Systematic glycemic control is the only procedure at our disposal to prevent diabetes long-term complications such as cardiovascular disorders, kidney diseases, nephropathy, neuropathy, and retinopathy. Glycated albumin (GA) has recently gained more and more attention as a control biomarker thanks to its shorter lifespan and wider reliability compared to glycated hemoglobin (HbA1c), currently the “gold standard” for diabetes screening and monitoring in clinics. Various techniques such as ion exchange, liquid or affinity-based chromatography and immunoassay can be employed to accurately measure GA levels in serum samples; nevertheless, due to the cost of the lab equipment and complexity of the procedures, these methods are not commonly available at clinical sites and are not suitable to home monitoring. The present review describes the most up-to-date advances in the field of glycemic control biomarkers, exploring in particular the GA with a special focus on the recent experimental analysis techniques, using enzymatic and affinity methods. Finally, analysis steps and fundamental reading technologies are integrated into a processing pipeline, paving the way for future point-of-care testing (POCT). In this view, we highlight how this setup might be employed outside a laboratory environment to reduce the time from measurement to clinical decision, and to provide diabetic patients with a brand-new set of tools for glycemic self-monitoring.

## 1. Introduction

According to the International Diabetes Federation, in 2021, 537 million adults were suffering from diabetes mellitus (DM), resulting in 6.7 million deaths and a 966 billion dollars healthcare cost [1]. DM is a worldwide-spread chronic metabolic disease that arises when the human body fails in the management of blood glucose levels. This condition can occur either because the pancreas does not produce enough of the hormone in charge of regulating glucose uptake, i.e., insulin, or because the cells cannot respond effectively to the insulin produced [2]. The former is referred to as type 1 diabetes, and is caused by an autoimmune response of the immune system that attacks the β-cells of the pancreas; what leads to this process is yet to be fully understood. It usually occurs in children and young adults, who experience polydipsia, polyphagia, and polyuria, and its management requires daily insulin injection to achieve the correct glycemic control, pivotal for patient long-term survival, and for reducing the risk of severe complications, especially at a vascular level [3]. The latter is known as type 2 diabetes and is the most common type of DM in the general population (more than 90% of diabetic subjects [1]), strongly connected to overweight and physical inactivity situations, as well as aging and ethnicity. Its symptoms are similar to type 1 DM but less accentuated, and it may sometimes present symptom-less, delaying the diagnosis and leading to complications, such as kidney diseases, nephropathy, neuropathy, and retinopathy [1]. Type 2 DM can be controlled with a healthy lifestyle, but pharmacological therapies, including oral hypoglycemic agents as well as insulin, are also available, intended to be used in the most severe cases [4]. Two other conditions have to be highlighted: (a) gestational diabetes, in which a hyperglycemic status arises during pregnancy, increasing the risk of adverse perinatal outcomes, including cesarean section, large for gestational age, and infant adiposity, and the risk of long-term pathological conditions for the mother (type 2 diabetes and cardiovascular disease) and their children (obesity and associated cardio-metabolic risks) [5,6]; (b) impaired glucose tolerance (IGT) and impaired fasting glycemia (IFG), in which blood glucose level is above the normal range but below the diabetes diagnostic threshold, hence being intermediate conditions between normality and diabetes [7].

Despite the existence of successful medical strategies to keep the DM condition under control, a definitive, gold standard therapy for treatment is still lacking, and patients remain lifelong at risk of relapse and uncontrolled disease [8]. The best current approach to prevent the aforementioned long-term complications, which contribute to an increase in the risk of death of the patient [9,10], is a prompt early diagnosis first, and a tight glycemic control afterwards. Table 1 summarizes the guidelines provided by the International Diabetes Federation [1], and reports the current thresholds adopted in clinics for the diagnosis of diabetes or pre-diabetes (i.e., IFG and IGT) conditions.

The two reference biomarkers for DM diagnosis and monitoring are blood glucose and glycated hemoglobin (HbA1c), an altered version of hemoglobin whose levels depend on the mean blood glucose in the previous months. However, they are not exempt from limitations: the former requires at least 8 h of dieting for the fasting plasma glucose (FPG) test, whereas the latter cannot be envisioned as a reliable marker in case the patient suffers from pathologies that alter the lifespan of erythrocytes, e.g., different types of anemia [11], or in case of hemodialysis [12,13]. Glycated albumin (GA) is another protein whose levels increase in hyperglycemic conditions and has recently gained more and more attention as a new biomarker thanks to a shorter lifespan compared to HbA1c, reflecting the mean blood glucose level over two to three weeks [14,15,16], thus being more responsive to rapidly evolving conditions. Moreover, it is unaffected by disorders that shorten the red blood cells lifetime, such as anemia, hence presenting itself as a better indicator whenever HbA1c measurements are not reliable. Various techniques such as ion exchange, liquid or affinity-based chromatography, and immunoassay can be employed to accurately measure GA levels in serum samples [17,18]; nevertheless, due to the cost of the lab equipment and complexity of the procedures, these methods are not commonly available at clinical sites and are not suitable to home monitoring. Two decades ago, an enzymatic method that uses liquid reagents with no preparation required has been developed, the Lucica^®^ GA-L by Asahi Kasei (Tokyo, Japan) [19], and it can be used with automated general biochemical analyzers providing clinicians with a faster and simpler procedure to measure GA in serum samples. More recently, innovative solutions in glycated albumin monitoring have been described, which evolved from the earlier enzymatic approaches, proposing alternative experimental affinity methods [20,21,22,23,24]. This new class of techniques has ignited unprecedented interest in point-of-care testing (POCT) devoted to DM monitoring. POCT, also known as “near-patient testing”, involves the use of portable devices to medically examine a patient during a consultation. Point-of-care devices can provide instantaneous results, supporting the benefit of timely patient care by better-informed healthcare professionals. Thanks to POCT, medical professionals are no longer required to wait for lab findings to elaborate an accurate diagnosis. Likewise, patients may save time and costs as they can operate the test at local healthcare providers or even directly at home. Point-of-care (POC) devices offer a wide range of advantages, including the provision of lab-quality results in minutes, effective near-patient diagnosis, less need for clinical visits, and more time to focus on treatment. Hatada et al. [25] detailed the current perspective of diabetes sensors implemented into POC devices, although their attention was strongly directed to the target molecules rather than to the technological deployment of the different biosensing mechanisms. To the best of our knowledge, a systematic analysis of the emerging technologies enabling GA monitoring is however still missing. The present review summarizes the most recent advances in the field of GA analysis, highlighting the main technological challenges to be faced to speed up the development of POC devices for the assessment of glycated albumin levels.

## 2. Glycemic Control Biomarkers

The current management of DM demands rigorous and systematic monitoring of the glycemic status of the patient, whose results can be exploited to assess therapy effectiveness, as well as to adjust diet and/or medications to improve blood glucose control. In the ’70s, diabetes was predominantly monitored through urine ketone and glucose measurements [26,27,28]. The former currently remains a valid tool to denote imminent or established diabetic ketoacidosis, i.e., a life-threatening complication induced by insulin deficiency, and hence it is recommended to all patients with diabetes, especially type 1 DM [29,30]. Conversely, technical advances in blood glucose monitoring (BGM) [31,32,33] together with additional clinical experience and extensive research investigations, marked the progressive abandonment of urine glucose testing as the recommended approach to home diabetes monitoring. Indeed, urine glucose level was proven to be an unreliable estimator of plasma glucose concentration due to wide confidence levels and poor correlation [34,35], significant inter-patient variability of glucose renal thresholds [36], and drug interference [37]. At the end of the last century, the results of two important randomized controlled trials were published. The Diabetes Control and Complications Trial (DCCT) [38,39] was conducted on 1441 type 1 diabetic patients, randomly assigned to an intervention group administered with intensive insulin therapy (three or more daily injections) guided by frequent BGM, and a control group in which patients followed conventional therapy requiring one or two daily injections; its findings cemented the clinical importance of rigorous monitoring aimed at maintaining the glycemic status of the subject as close as possible to the normal range. The UK Prospective Diabetes Study (UKPDS) [40] involved 3867 type 2 diabetic patients randomly split into an intervention group receiving different sulfonylureas (i.e., chlorpropamide or glibenclamide) or insulin and a control group treated with conventional diet; the study aimed at establishing whether intensive glucose control had an impact on lowering the risk of macro/micro-vascular complications, and whether any pharmaceutical therapy was more advantageous than the others. Its results showed that improvement in glycemic control, assessed over a 10-year temporal window with HbA1c systematic monitoring, rather than any specific therapy, was the principal factor involved in the observed risk reductions. As a consequence, nowadays, glycemic control is assessed in clinics by the HbA1c measurement, whereas continuous glucose monitoring (CGM) and BGM are useful individual tools for diabetes self-management [41].

### 2.1. Glycated Proteins

Blood glucose and urine ketone measurements are single-point measurements that provide essential information on diabetes management on a daily basis; glycated proteins instead, such as HbA1c and GA, introduce a new, complementary layer in glycemic control monitoring by reflecting the mean glucose level over longer, past periods, and are not affected by daily fluctuations induced by diet or physical activity. Glycation is a non-enzymatic mechanism also called a Maillard reaction, which consists of the covalent addition of a reducing sugar to a free amino group of amine-containing molecules such as proteins to form an unstable, reversible product (i.e., Schiff base) which is then rearranged to a more stable conformation known as Amadori product or ketoamine. This process is shown in Figure 1.

Eventually, this process leads to the formation of irreversible compounds designated as advanced glycation end-products (AGEs) [42]. Advanced glycation is a critical pathway involved in the development of several diabetic complications such as neuropathy, nephropathy, and retinopathy that arise from AGEs-induced oxidative stress and inflammatory processes [43]. Protein glycation is affected by the time of exposure to glucose, and its concentration; extracellular proteins such as albumin have higher glycation rates than intracellular ones, such as hemoglobin, due to their direct exposure to blood glucose [44].

#### 2.1.1. Glycated Hemoglobin

HbA1c is the Amadori rearrangement of the adduct of glucose with the N-terminal valine of the β-chain of hemoglobin [29,45,46], which is the most reactive site [47,48]. Its rate of formation is proportional to the ambient glucose concentration, and it reflects the mean glycemia over the past two to four months, correlating directly with the lifespan of erythrocytes [29,49,50,51]. Its value is expressed in terms of percentage with respect to the total hemoglobin concentration and can be used as a diagnostic biomarker [1] and as a monitoring tool to assess treatment effectiveness in diabetic patients [41]. Despite being supported by large-scale clinical trials, i.e., the DCCT and the UKPDS, its employment suffers from some intrinsic disadvantages related to the breadth of the temporal window, which does not allow for accurately tracking rapid changes in glycemic control [52,53,54], and to its reliability under certain clinical circumstances such as hematologic disorders (variant hemoglobin, different types of anemia), recent blood transfusions, use of erythropoietin-based drugs, and pregnancy, which alter the lifespan of red blood cells hence affecting HbA1c measurements [16,41,55]. Moreover, there is evidence for inter-individual heterogeneity of glucose gradient across the membrane of red blood cells, which changes the dynamics of hemoglobin glycation hence impacting HbA1c assessment tests [56].

#### 2.1.2. Glycated Albumin

Human serum albumin (HSA) is the most abundant protein in human blood: with a normal concentration ranging from 3 to 5 g/dL, it accounts for approximately 60% of serum proteins [44,57]. It is composed of a single, 585 amino acids-long polypeptidic chain with a molecular weight of 66.5 kDa [58,59], and its three-dimensional structure is reported in Figure 2. HSA has a half-life of approximately three weeks [60], during which the exposure to blood glucose induces glycation processes primarily at its lysine and arginine residues [15] that modify its spatial arrangement as well as the N-terminal region [16]; glycation of albumin also leads to a slight increase in the polarity of the molecule [46].

Clinically, GA has some clear advantages over HbA1c. Firstly, thanks to a higher rate of formation and shorter lifespan, it can reflect hyperglycemia earlier than HbA1c [16], and it is a more adequate indicator to evaluate glycemic variability [53,61]. Secondly, due to its independence from red blood cells, it offers a more robust parameter whenever the patient suffers from erythrocyte lifespan-affecting events. Table 2 summarizes the principal clinical conditions in which GA may offer a better understanding of the glycemic status of a patient.

GA, however, is not exempt from limitations, and medical operators should be aware of the conditions in which glycated albumin does not accurately reflect the glycemic status of a patient because of the involvement of other factors. GA measurements, being corrected for total albumin, should not be influenced by albumin concentration [44,70]; nevertheless, the association between low plasma albumin levels and increased protein glycation rates, probably caused by different exposure to glucose, has been demonstrated [71]. Indeed, disorders that impact HSA metabolism may alter GA levels; in particular, higher GA levels have been observed in patients with chronic liver disease [72] and hypothyroidism [73], whereas lower values have been reported for nephrotic syndrome [74,75] and hyperthyroidism [73] cases. Body mass index (BMI) has an impact on glycated albumin too [76], with absolute GA values decreasing by 0.13% every 1 kg/m^2^ increment in BMI [77]. Finally, patient age should be considered when analyzing GA levels, since newborns show much lower values of GA with respect to adults [78], and the values significantly increase as the patient’s age increases [79,80]. According to the Japanese Diabetes Society, values of GA in non-diabetic patients should range within 11–16%, normalized to the total albumin, whereas diabetic patients generally exhibit values greater than 20% [81]. Nevertheless, the lack of standardization in the reference method used to assess diabetes (some studies used FPG, others oral glucose tolerance test (OGTT) or HbA1c) is responsible for slight variations in the definition of the reference thresholds. Another critical aspect is related to the choice of the analytical technique used to obtain the GA measurement. Kohzuma et al. [45] meticulously summarized the main clinical studies and their relative findings related to GA reference range and cutoff values for diabetes diagnosis and screening, whereas Roohk et al. [82] reported the reference values employed in six US clinical laboratories, showing the discrepancies related to the different GA testing methods used.

## 3. Glycated Albumin Analysis

### 3.1. Laboratory Techniques

Traditional methods for glycated proteins’ detection include colorimetric assessments with thiobarbituric acid assay or nitroblue tetrazolium test [83,84,85,86,87], chromatography with anion exchange to isolate albumin and subsequent boronate affinity to distinguish between glycated and non-glycated versions [88,89,90,91,92], and immunoassay techniques such as enzyme-linked immunosorbent assay (ELISA) and enzyme-linked boronate-immunoassay (ELBIA) [18,93]. In order to avoid expensive, voluminous laboratory equipment and/or time-consuming procedures, GA is currently measured with enzymatic kits at clinical sites. This technology relies upon fructosamine oxidase (FAOX), which can be applied to automated biochemical analyzers and require no prior preparation of samples. The two commercially available kits are the Lucica^®^ GA-L by Asahi Kasei (Tokyo, Japan) [19], whose performances and clinical utility have been validated by the Italian teams of Paroni et al. [94] and Testa et al. [95], and its European version produced and commercialized as quantILab^®^ Glycated Albumin by Instrumentation Laboratory SpA—Werfen (Milan, Italy), for which Paleari et al. [96] have performed the first multi-center evaluation to assess its clinical suitability.

### 3.2. Experimental Techniques

To minimize the time elapsed between a pathological variation in DM biomarkers and the corresponding intervention of the clinician, resulting in prompt medical assistance, and to provide the patients with more tools for diabetes self-care and monitoring, recent research has focused on the development of point-of-care testing solutions that can be easily and quickly employed outside a laboratory environment [25]. Currently, no POC solution is commercially available, but several proposals of biosensors able to serve this purpose have been developed in recent years. This section will offer an overview of the most innovative and promising advances related to GA point-of-care analysis; a table, inserted at the end of this section, will summarize the performances of all the discussed solutions.

To better understand if a method is suitable for GA levels monitoring, the following reasoning could be applied. Taking as a reference the values reported by the Japanese Diabetes Society, i.e., 11–16% for non-diabetic and >20% for diabetic patients [81], and considering a HSA concentration of 5 g/dL, a limit of detection (LOD) for GA of at least 0.2 g/dL (30 μM) is needed to distinguish between normal (worst case, i.e., 16%) and pathological condition. Under the same hypothesis, to be able to measure a single-step percentage variation, the LOD should be at least 0.05 g/dL (7.5 μM).

#### 3.2.1. Enzymatic Methods

In 2005, Yamaguchi et al. [97] proposed a dry chemistry system that exploits an enzymatic method, based on a test strip with three separated zones (called *test-tapes*) sensitive to glycated albumin, albumin, and ketoamine, respectively, so that the final GA value can be expressed in percentage with respect to total albumin, and corrected for blood ketoamine presence. In each zone, a specific chain of reactions induces a colorimetric change that can be analyzed at a particular wavelength by a portable optical analyzer. After the calibration of each test-tape, the authors retrieved a final expression of the GA value that showed a correlation coefficient *R* of 0.82, with a coefficient of variation CV less than 10% within the 9.6 to 14% range, but with an increase up to 20% in the 17 to 20% range of GA values. Some of the problems related to this solution are the necessity to heat the three zones differently (the GA test-tape works better at 37 °C), and small sample evaporation during the measurement due to light exposure. More recently, enzymatic methods for POC tests have shifted towards electrochemical sensing solutions, which enable the development of cheap, small, and easily embeddable sensors typically based on screen-printed carbon electrodes (SPCEs) or interdigitated electrodes (IDEs). Depending greatly upon the type of substrate required by the enzyme, there is a distinction between methods that need a pre-digestion of the sample and methods that can interact directly with intact glycated albumin.

In 2017, Hatada et al. [98] developed an SPCE-disposable electrochemical enzyme sensor strip based on the GA measuring principle of the Lucica^®^ kit: initially, the sample is pre-treated with a protease to release ϵ-fructosyl lysine (ϵ-FK) from GA; this substrate is then oxidized by FAOX deposited on the SPCE. To retrieve an electrical signal, an electron mediator is used: a ruthenium complex (hexaammineruthenium(III) chloride) accepts the electrons involved in the oxidation of ϵ-FK, and is simultaneously reduced; then, by applying an oxidative potential to the counter electrode of the sensor, the mediator is oxidized back and the electrons collected at the working electrode generate a current that can be related to the concentration of ϵ-FK present in the sample. In 2021, the same authors [99] modified the design of their biosensor by substituting the SPCE with an IDE, which highly ameliorated the sensitivity and reproducibility of the measurement by providing a higher magnitude of current and further reducing the time needed to reach a steady-state condition, which occurred almost instantaneously. The LOD for the synthetic version of the ϵ-FK (i.e., Z-FK) dropped from 40 μM to 1.2 μM and the sensitivity, previously 0.49 nA/μM, reached 2.8 nA/μM. Figure 3 provides a schematic representation of the electrochemical enzymatic analysis of GA.

The enzymatic biosensors based on FAOX are highly sensitive, repeatable, and stable; nevertheless, the long time required by the proteolytic digestion stands as the main limitation of these solutions, possibly affecting their diffusion in a POC regime. To overcome this issue, procedures that require no pre-digestion have been investigated. Kameya et al. [100] exploited an enzyme retrieved from *E. coli*, fructosamine 6-kinase (FN6K) that can interact with intact GA; however, as pointed out by the authors, the reactions involved in this design require several additional steps and reagents (such as enzymes, co-substrates, buffers, mediators) that increase the complexity of the manufacturing and may lead to errors in the measurements. Table 3 compares the technologies presented.

#### 3.2.2. Affinity Methods

Several biosensors exploiting recognition molecules in order to detect GA in samples will be presented. In Figure 4, the different techniques here described have been conceptualized and represented in a simple, but functional visual manner.

Immunosensors, specifically electrochemical immunosensors, detect a variation in the electrical properties of the sensing element upon antibody/antigen complex formation. In 2017, Bohli et al. [20] worked on an IDE-based immunosensor in which the electrode surface has been functionalized with the immobilization through physisorption of anti-HSA monoclonal antibodies. They used cyclic voltammetry to characterize the steps of the functionalization, and eventually, electrochemical impedance spectroscopy (EIS) was employed to detect and quantify changes in the impedance of the sensor due to GA/antibody binding events; in particular, the authors noticed a decrease in the charge transfer resistance of the system possibly caused by rearrangements in the monoclonal antibody structure upon antigen recognition, which changes the conductive properties of the sensor and could be correlated to GA concentration through an exponential relationship.

Aptamers are three-dimensional oligonucleotides (RNA or single-stranded DNA structures) that can bind with high selectivity and specificity to target molecules thanks to their spatial conformation [101]. They offer critical advantages with respect to antibodies: (a) protein-based antibodies irreversibly denature at high temperatures, whereas oligonucleotides offer greater thermal stability which extends their shelf-life; moreover, their conformation–recovery properties potentially make aptamer-based sensors recyclable [102]; (b) antibodies require complex and expensive processes for their fabrication and activity control, whereas elevated quantities of aptamers can be easily synthesized (also in modified forms) in highly reproducible and cost-effective manners [103]; (c) aptamers are identified and produced with *in vitro* processes that do not involve animal cells; for this reason, their properties can be changed on demand, and they can be synthesized to recognize a wider variety of ligands such as small molecules, ions, and proteins [104]. These properties make aptamers strong candidates for the development of GA biosensors, especially in a POC context. In 2019, Bunyarataphan et al. [21] developed an electrochemical aptasensor based on two streptavidin-modified SPCEs functionalized with GA and HSA-specific aptamers, respectively; both aptamers have been modified at their 5′ terminal with biotin to bind to streptavidin: this strategy increases the number of aptamers attached on the surface and, most importantly, orders their spatial arrangement facilitating the bindings with the target ligand. The SPCEs were pre-treated with the application of an anodic potential in an acidic environment, after which streptavidin first and aptamers then have been deposited on the surface of the electrodes; each step of the functionalization has been verified with cyclic voltammetry, whereas square wave voltammetry (SWV) was employed as a measuring technique. In a ferricyanide solution, the formation of aptamer/ligand complex hinders the transfer of electrons involved in the Fe^2+^/Fe^3+^ redox reaction at the sensing surface level, inducing a drop in the current collected at the working electrode. The authors managed to find a linear relationship between current drop and target concentrations, with a coefficient of determination $R^{2}$ of 0.989 and 0.994 for GA and HSA sensors, respectively. The LOD was 2.6 ng/mL for the GA aptasensor and 0.2 μg/mL for the HSA aptasensor, and the final GA value is expressed as a percentage with respect to total albumin. Farzadfard et al. [22] in 2020 developed an aptasensor in which the working electrode (a glassy carbon electrode) of an electrochemical cell has been treated with reduced graphene oxide (rGO) sheets and gold nanoparticles to provide a large surface area and enhance the sensitivity of the electrode. The working principle of the sensor is similar to the previous one: a solution of potassium hexacyanoferrate(II) is used to create a Fe^2+^/Fe^3+^ redox couple; the binding of GA to the aptamer complex hampers the electron transfer at the surface whenever a voltage potential is applied to the working electrode, thus a decrease in the signal current proportional to the amount of ligand present can be detected. Functionalization steps have been studied with both cyclic voltammetry and EIS. As a measurement technique, EIS has given the best results: a linear relationship ($R^{2}\;=\;0.997$) has been found between the charge transfer resistance of the electrode and the concentration of glycated albumin in the sample, with an LOD of 0.09 mg/mL. The authors also developed a non-toxic version of this sensor that does not require potassium ferrocyanide to produce an electrochemical signal, but exploits a methylene blue-modified aptamer; they investigated its performances with SWV and obtained an LOD of 0.07 μg/mL, but the calibration curve showed a double slope profile. In 2021, Waiwinya et al. [23] designed an immobilization-free aptasensor for GA detection that required no functionalization of the SPCE surface other than a simple pre-treatment with ethanolamine and a phosphate-buffered saline wash. The sample to be analyzed was added to a reaction mixture containing free graphene oxide (GO) sheets with aptamers adsorbed on their surface through π−π interactions, and potassium ferricyanide; this solution was then dropped onto the surface of the pre-treated SPCE. The aptamers detached from GO sheets to bind to the target, leading to the deposition of free GO on the electrode surface and a subsequent enhancement of the electrochemical signal proportional to the concentration of GA in the sample. The authors found a linear correlation ($R^{2}\;=\;0.989$) between current variations with respect to bare electrode conditions (measured with SWV) and base-10 logarithmic GA concentration; the LOD was 8.70 ng/mL. The authors tested their design also with a fluorescent version of the sensor to confirm the interaction between the selected aptamer and GA. In addition to a very low LOD, this sensor showed an extremely fast assay time (30 min) because no immobilization was required, making it a good candidate for a POC scenario. A different approach has been followed by Sasar et al. [101] in 2020, where the authors worked on a biosensor based on a modified field-effect transistor (FET); this technology offers the advantages of complementary metal-oxide semiconductor (CMOS) compatibility, miniaturization, low power consumption, and could also be realized on flexible substrates. In this design, the gate of the transistor has been covered with an insulating layer of silicon dioxide (SiO_2_) on which randomly oriented, gold-coated zinc oxide (ZnO) nanorods have been hydrothermally deposited. The gold layer offers an anchor point for the thiol group of the modified aptamers employed as recognition elements, and its interaction with ZnO induces a negatively charged layer at the SiO_2_ surface that decreases the conductivity of the p-type silicon channel of the FET. The attachment of GA to the aptamers further accentuated the presence of negative charges in the silicon channel, and the subsequent drop in conductivity has been related by the authors to the GA concentration in the sample under analysis. No data concerning the construction of a calibration curve or a LOD have been presented for this technology, but the authors reported a very fast (in the order of minutes) response time implying real-time application possibilities. Table 4 summarizes the nucleotide sequences of the aptamers employed in the works described above.

The last affinity solution here presented is taken from an article published in 2019 by Attar et al. [24]. The authors described a polymer-based electrode able to detect and quantify both HSA and GA in a single sample. The design involves a particular green fluorescent protein (GFP) construct with affinity for glycated and non-glycated albumin, immobilized on the surface of a custom-made electrode. Specifically, the working electrode has been coated with a poly-ethylenedioxythiophene (PEDOT) film bearing an iminodiacetic acid motif, and the binding with the GFP recognition element (in the article referred to as *α-HSA*) occurred via copper(II). The PEDOT/copper/α-HSA/target complex formation generates an EIS signal that allows for quantifying the total HSA + GA concentration. The quantification of GA alone is made instead through an engineered variant of the dihydrofolate reductase (DHFR) enzyme that targets selectively the carbohydrate elements of GA: under an applied potential that oxidizes the product of the enzymatic reaction, a current can be collected at the working electrode through SWV. In this way, the authors were able to measure glycation ratios of HSA in a sample between 5 and 80%, even if the relationship with current turned out to be nonlinear. Table 5 compares the technologies presented.

#### 3.2.3. Additional Relevant Approaches

It is also worth mentioning that additional works that can be found in the literature because even if not fully compatible with an ideal POC regime yet, they still provide relevant insights into the technologies available nowadays. Amongst the several proposals of aptasensors, in 2016, Apiwat et al. [105] developed an ELISA-like assay that exploits the fluorescent quenching interactions between GO and a cyanine (Cy5)-labeled aptamer. When the single-stranded DNA structures form π−π binding with graphene oxide, the fluorescent signal is quenched; in the presence of GA, a certain amount of aptamer detaches from GO to bind to the target analyte, resulting in a recovery of the fluorescent signal proportional to the GA concentration. The authors discovered a sigmoidal relationship between the fluorescent signal and the GA concentration, with the linear region ($R^{2}\;=\;0.98$) being between 0.05 and 0.3 mg/mL, thus requiring at least a 4-fold dilution of the sample; nevertheless, the method showed good performances, with an LOD of 50 μg/mL. Belsare et al. [106] in 2021 took advantage of aptamer conjugation to gold nanoparticles to eliminate the need for dyes and simplify signal generation and processing, and developed a paper fluidic dipstick assay for the colorimetric measurement of GA in gestational diabetic patients. The assay is made of a dipstick strip on which a capture zone, with a general albumin aptamer immobilized, has been created; depending on the target analyte to be quantified (GA or HSA); the sample is mixed with a solution containing GA or HSA-specific aptamers/gold nanoparticle complexes that, once attached to the corresponding albumin version trapped inside the capture zone, produce a colorimetric signal that can be optically analyzed and related to target concentration. The authors were able to measure both GA and HSA within their physiological concentration ranges: 50–300 μM (0.35–2 g/dL) with an LOD of 8.7 μM for GA in bovine serum without dilution and 500–750 μM (3.5–5 g/dL) with an LOD of 24 μM for HSA in bovine serum without dilution. A different approach has been followed in 2020 by Ki et al. [107]: they developed a two-strip advanced lateral flow immunoassay (LFIA) sensor to retrieve information on albumin glycation ratios by parallel, accurate measurements of total HSA and GA concentrations via colorimetric analysis. In a conventional LFIA strip, the liquid sample deposited on a *sample pad* runs together with a loading solution through a series of capillaries until it reaches a *conjugate pad*, where a mixture of recognition particles is stored. The target molecules in the sample bind to these particles and are later trapped inside two lines, test and control, where a color signal is generated. The innovative design, instead, includes a disconnected bridge structure with the conjugate pad containing anti-HSA antibodies/gold nanoparticle complexes placed after the loading solution inlet pad and before the sample inlet pad. The separation allows the sample solution to react alone with the specific antibodies in the capture zones; later, when the loading solution transporting the complexes arrives on the binding site, these structures attach only to the correct version of the protein already anchored. In this way, the higher concentration of HSA in the sample, together with the higher affinity of the complexes towards HSA with respect to GA, do not bias the measurement. In this way, the authors managed to remove HSA interferences and obtained a much higher intensity signal with respect to conventional LFIA, resulting in the determination of physiological glycation ratios over a wide range of GA and total HSA concentrations. The authors are working on improvements to this design to remove the need for multiple injections of solutions.

Instead of relying on antibodies or aptamers, Paria et al. [108] in 2021 published a paper describing a sensor based on surface-enhanced Raman spectroscopy (SERS)-active substrates able to perform label-free detection of GA. Specifically, they adopted randomly oriented silver-coated silicon nanowires (Ag/SiNWs) that, when the sample deposited on them is left evaporating, cluster around target molecules and trap them, leading to the formation of so-called *plasmonic hotspots*. By analyzing the Raman spectra within these regions, the authors were able to exquisitely distinguish between GA concentrations with a LOD of 500 nM. In addition, they collected SERS spectra from samples with 5 to 25% glycation ratios and build a chemometric classifier to quantify GA levels in GA/HSA mixtures. Due to the extremely low detection limit of this sensor, it could be employed also in non-invasive scenarios such as saliva and tears analysis.

The main limitation that could represent an obstacle to the diffusion of these biosensors in a POCT scenario is the need to perform optical or spectroscopic techniques for the analysis of the fluorescent, colorimetric or spectroscopic signals, which require laboratory technologies. To address this problem, in recent years, proposals for smartphone-oriented hardware and/or software solutions have been investigated [109,110,111,112,113]. Uncontrolled or variable lighting conditions, inconsistencies between different smartphone models (especially tone-related), and possible hardware/software auto-corrections implemented in smartphone camera applications, however, are still issues to be fully tackled [114,115]; in addition, biosensors should be designed taking into consideration the final readout technology for better compatibility between the sensing element and the smartphone-based solution [116]. Table 6 compares the last four technologies presented, whereas Table 7 provides a synthesis of the performances of all the biosensors analyzed in this review.

## 4. Operational and Technological Challenges towards Innovative Point-of-Care Testing for GA Detection

GA point-of-care testing can promote operational efficiency. Significant improvement of glycemic control can be reached (a) in cases of intensive insulin therapy, where frequent monitoring of blood glucose is required [117]: self-assessment of GA can provide the patient with evidence that glucose measurements were reliable, and the therapy is being followed appropriately; (b) in case of life-threatening and difficult to manage conditions such as fulminant type 1 DM, where a prompt diagnosis is of crucial importance due to the rapidly evolving nature of the disease: a GA cut-off value of 33.5% has been found as capable of distinguishing between type 1 and fulminant type 1 DM [118]; (c) in cases of anemia, genetic variants of hemoglobin and chronic kidney disease: the impact of a GA point-of-care testing solution lies in the reliability of GA with respect to HbA1c as a biomarker coupled to the already discussed advantages of a POC device; d) in cases of pregnant women: the glycemic control monitoring with GA is superior with respect to HbA1c because of a better stability of the biomarker and compatibility with gestational duration, and the POC solution can provide the patient with a tool that can be used at home without causing any mobility-related stress.

A simplified analysis pipeline for a POC device, consisting of the main functional steps, namely blood collection, plasma extraction, GA processing, electrical transduction, and finally GA reading, was exemplified (Figure 5—upper chart). These steps were mapped into operational and technological solutions (Figure 5—lower chart), corresponding respectively to finger pricking, microfluidic-based plasma separation by passive/active capillarity, enzymatic/affinity methods to process the GA, employment of electrodes/transistors to transduce the electrochemical reactions into an analytical signal, and finally the electronic circuitry to perform the reading. Some challenges can be identified at each level: (a) blood collection via finger pricking could be acceptable in a risk-benefit view but remains an invasive procedure. Some studies tried to measure GA in salivary samples [119,120], but there is still a gap within the scientific knowledge on the correlation between changes in salivary fructosamines content and fluctuations of serum biomarkers [121]; (b) the separation of blood parts to extract serum/plasma is a critical phase during which the occurrence of hemolysis could alter the following measurement, especially in enzymatic methods; (c) the choice of the most suitable analysis method has to be done evaluating the strengths and weaknesses of each solution. Enzymatic biosensors are a well-established technology that can take advantage of previous knowledge (for instance, glucose biosensing strips), whereas sensors based on aptamers, despite showing great performances, are a relatively new technology and a variety of solutions is present (each work introduces a different aptamer sequence, or even a different recognition element, coupled to a different substrate exploiting a different electrochemical transduction mechanism), hampering the understanding of which is best. Another aspect that needs to be considered is the time required by each specific procedure: enzymatic methods, especially those relying on the proteolytic digestion of the sample, are not particularly efficient, whereas methods that can interact with intact GA are faster; and (d) electrical transducers are present under the form of electrodes (SPCEs or IDEs) or transistors technology, which offer great performances in terms of scalability, but their application-oriented fabrication can increase the cost of the biosensor, especially if intended to be disposable. These biosensors, indeed, not only contain precious metals, but their preparation requires high-end manufacturing processes and expensive materials; (e) eventually, the low target quantities to be analyzed demand high-end circuitry able to provide enough sensitivity to distinguish between the different concentrations.

Affinity aptamer-based methods are promising candidates for a POC platform. The properties of aptamers make these biosensors flexible in design while conferring them stability under various conditions, especially compared to enzymatic and antibody-based ones which have more stringent temperature requirements, thus facilitating their storage and extending their shelf-life. Aptamers can be selective for intact GA, lowering the assay time, and the working principle of these solutions can be investigated with simple electrochemical measurement techniques that are not only fast but also embeddable in portable devices. In addition, the literature offers examples of aptamer-based sensors that require no functionalization of the transducer element (such as the work from Waiwinya et al. [23]), hence limiting both the costs of production and possible reproducibility errors caused by additional, complex manufacturing steps.

## 5. Conclusions

Given the chronic nature of the DM, along with the induced devastating systemic complications, POC-based scenarios for glycemic biomarker assessment are to be envisioned for early detection and disease development monitoring. In this view, POC devices will enable biomarker measurement into an integrated processing pipeline, which might be implemented in a home-based setup to facilitate DM management by the patients and clinicians. Alongside the traditional blood glucose and glycated hemoglobin biomarkers, glycated albumin has been the subject of intense studies lately. While evidence of GA benefits in clinical routine is rapidly accumulating, more investigation is still needed to determine how best to harness GA properties, against other glycemic control biomarkers, and the available analytical techniques. In this review, we revised the recent trends in GA analysis entailing two main biochemical techniques, which leverage enzymatic and affinity methods, respectively. Technological and operational challenges of GA-based point-of-care devices were also highlighted, underlying that a transition from optical to electrochemical reading techniques is fundamental to ensure device compactness and cost savings. In conclusion, we may assert that:adding GA to traditional glycemic control markers may contribute to improving overall DM monitoring;aptamer-based sensors are expected to overcome some issues typical of enzymatic-based sensors, but reading technologies still need to be systematically verified;GA-based POC devices are promising and may revolutionize remote glycemic monitoring.

## Figures and Tables

**Figure 1 biosensors-12-00687-f001:**
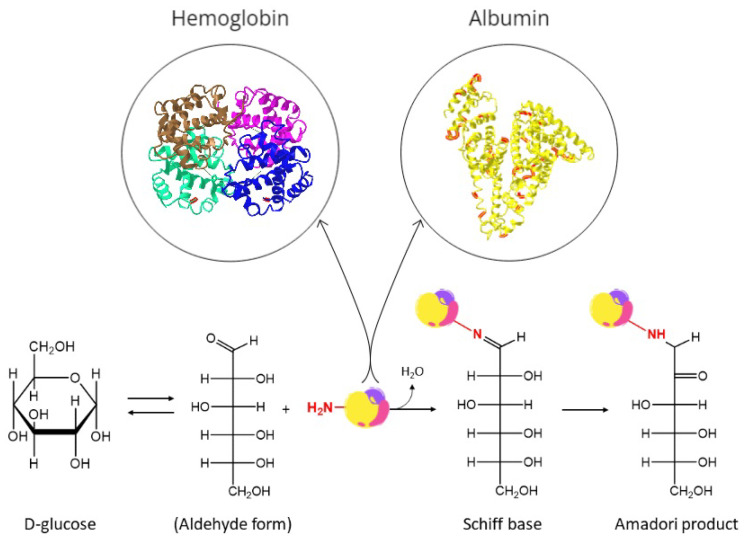
Reactions involved in the glycation of proteins. In particular, hemoglobin (PDB ID: 1BBB) and albumin (PDB ID: 1AO6) have been reported, and the main glycation sites for each of them have been highlighted in red: N-terminal valine of the β-chains of hemoglobin, and lysine and arginine residues of albumin.

**Figure 2 biosensors-12-00687-f002:**
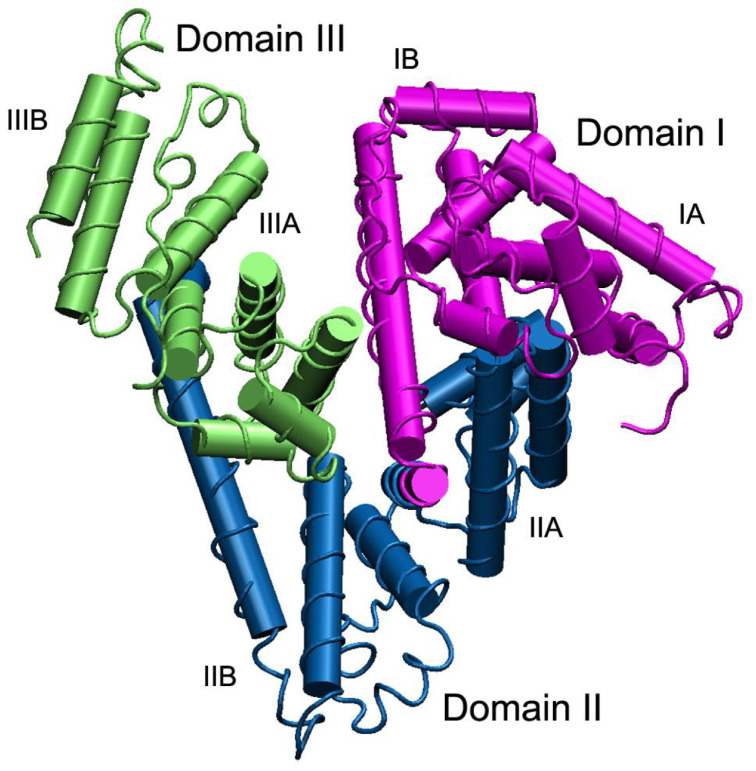
Three-dimensional structure of human serum albumin. The three domains I, II, and III are highlighted in purple, blue and green, respectively, and for each domain the two subdomains A and B are shown—from Belinskaia et al. [59].

**Figure 3 biosensors-12-00687-f003:**
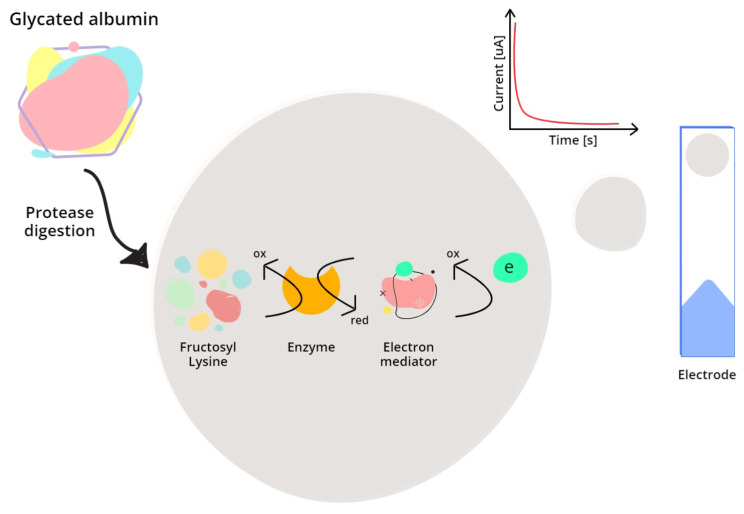
Schematic process of an electrochemical enzymatic analysis of glycated albumin. Initially, the sample has to undergo a proteolytic digestion to release ϵ-FK from GA. Then, a specific enzyme (FAOX) oxidizes this substrate while simultaneously an electron mediator is reduced. Finally, by applying an appropriate voltage potential at the electrode site, the reduced-form mediator is oxidized back, releasing electrons that can be collected to measure a current.

**Figure 4 biosensors-12-00687-f004:**
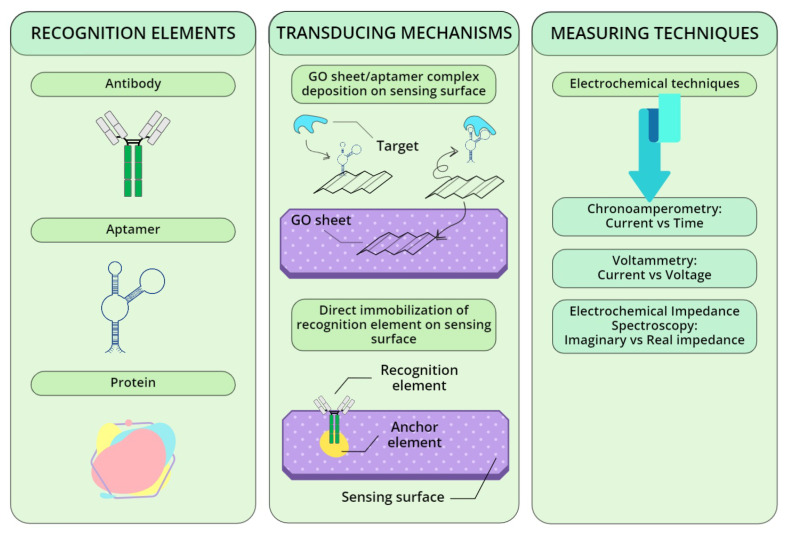
Analysis of the three main aspects of a biosensor realization (i.e., choice of the recognition element; transducer design implementation; analytical measuring technique) applied to POC-compatible affinity methods.

**Figure 5 biosensors-12-00687-f005:**
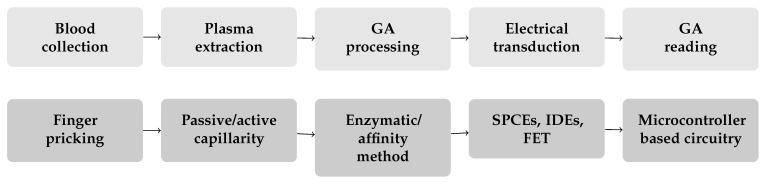
Schematic representation of the hypothesized steps involved in a POC testing device. The actions to be performed are on top, whereas, at the bottom, the solutions that could be adopted to fulfill the respective needs.

**Table 1 biosensors-12-00687-t001:** Threshold levels for the diagnosis of diabetes or pre-diabetes according to the International Diabetes Federation.

Condition	Main Criterion	Alternative Criteria
Diabetes	FPG ^1^ ≥ 126 mg/mL	OGTT ^2^ ≥ 200 mg/mL or HbA1c ^3^ ≥ 6.5%
IGT ^4^	FPG ^1^ < 126 mg/mL with OGTT ^2^ 140–200 mg/mL	
IFG ^5^	FPG ^1^ 110–125 mg/mL with OGTT ^2^ < 140 mg/mL	

^1^ Fasting plasma glucose, ^2^ Oral glucose tolerance test, ^3^ Glycated hemoglobin, ^4^ Impaired glucose tolerance, ^5^ Impaired fasting glycemia.

**Table 2 biosensors-12-00687-t002:** List of conditions in which GA may be more reliable than HbA1c as a glycemic control biomarker.

Condition	Brief Explanation	Reference
Intensive insulin therapy	HbA1c levels change too slowly, whereas GA tracks accurately the variations induced by the therapy.	[19]
Fulminant type 1 diabetes mellitus	HbA1c is usually normal or only slightly elevated in this clinical condition, whereas the GA/HbA1c ratio significantly increases due to the reactiveness of GA. In this case, the GA/HbA1c is even a better indicator than GA alone.	[62,63]
Anemia	*Hemolytic* anemia shortens the lifespan of erythrocytes hence HbA1c levels are lower, whereas *iron deficiency* anemia produces higher levels of HbA1c due to enhanced glycation processes and longer red blood cell survivability. Albumin instead is not affected by these pathologies.	[64,65]
Variant hemoglobin	The genetic structural variants affect the ability of hemoglobin to be glycated, hence HbA1c does not reflect properly the glycemic status. Albumin instead is not affected by this condition.	[66]
Pregnancy	Towards the end of pregnancy, iron deficiency affects HbA1c levels, whereas GA levels remain stable both in diabetic and non diabetic women.	[67,68]
Chronic kidney disease	Erythropoietin injections, blood transfusions and hemodialysis are frequent in patients with this condition. They all affect red blood cells’ lifespan and iron values, altering HbA1c levels. Special attention has to be put on proteinuria conditions that may develop in these patients because GA values can be altered.	[12,13,69]

**Table 3 biosensors-12-00687-t003:** Enzymatic biosensors comparison. The ‘Complexity’ column indicates qualitatively the difficulty of the manufacturing process for a specific solution. The ‘Deployment phase’ column indicates at which level the technology is actually adopted: ‘Laboratory’ suggests an ad-hoc experimental phase, ‘Pre-clinical’ specifies that the biosensor has been tested on real patients samples in a controlled environment.

Detection Mechanism	Advantages	Disadvantages	Complexity	Deployment Phase	Ref.
Three different enzymatic reactions on three different strips to analyze GA, total albumin, and ketoamine. Colorimetric evaluation of each strip	Measures both GA and HSA; accounts for ketoamine interference	Bulky; need for temperature control; possible sample evaporation	High	Laboratory	[97]
Electrochemical reaction involving an enzyme, FAOX, and ϵ-FK as a substrate. Electron transfer is mediated by a ruthenium complex and current vs. time is measured	Very fast measurement (1 min); embeddable	Time-consuming pre-digestion of the sample (long assay time); possible lot-to-lot variations	Moderate	Laboratory	[99]
Electrochemical reaction involving an enzyme, FN6K, and intact albumin as a substrate. Electron transfer is mediated by a methylsulfate element and current vs. time is measured	No digestion required; fast assay time (10 min); embeddable	Complex chain of reactions and presence of many solutes with different solubility are important sources of errors	High	Laboratory	[100]

**Table 4 biosensors-12-00687-t004:** Nucleotide sequences of the aptamers adopted in different works to detect GA and/or HSA, with respective modifications and target molecule they are designed to bind to.

Aptamer Sequence 5′-3′	Modifications	Target	Reference
TGCGGTTGTAGTACTCGTGGCCG	Biotin at 5′	GA	[21]
H8 aptamer ^1^		HSA	
GGTGGCTGGAGGGGGCGCGAACGTTTTTTTTTT	Thiol group at 3′ and methylene blue at 5′	GA	[22]
TGCGGTTCGTGCGGTTGTAGTAC	Unmodified	GA	[23]
TGCGGTTCGTGCGGTTGTAGTAC	Fluorescein at 5′	GA	
GGTGGCTGGAGGGGGCGCGAACGTTTTTTTTTT	Thiol group at 3′	GA	[101]

^1^ ATACCAGCTTATTCAATTCCCCCGGCTTTGGTTTAGAGGTAGTTGCTCATTACTTGTACGCTCCGGATGAGATAGTAAGTGCAATCT.

**Table 5 biosensors-12-00687-t005:** Affinity biosensors comparison.

Detection Mechanism	Advantages	Disadvantages	Complexity	Deployment Phase	Ref.
Anti-HSA monoclonal antibodies have been anchored to an electrode surface. The binding with the antigen (HSA or GA) changes the impedance of the sensor	Relatively fast assay time (15 min); embeddable	Possible cross-interference between HSA and GA, both present in a real sample, due to a-specificity of recognition element	Moderate	Laboratory	[20]
GA and HSA-specific aptamers immobilized on two different SPCEs. Under applied potential, electron transfer in ferricyanide solution is hindered by aptamer/ligand complex formation and current vs. voltage is measured	Elevated stability (4 weeks); embeddable	Long assay time (40 min)	Moderate	Pre-clinical	[21]
rGO and gold nanoparticles-treated electrode surface with methylene blue-modified, GA-specific aptamers immobilized. Under applied potential, electron transfer is hindered by aptamer/ligand complex formation and current vs. voltage is measured	Elimination of electron mediator through modified aptamer	Embeddability: conventional electrochemical cell was used instead of disposable electrodes; complex measurement procedure	High	Laboratory	[22]
In the presence of GA, aptamers previously attached to GO sheets in a solution selectively detach from the sheets and bind to the target. Deposition of free GO sheets onto the electrode surface cause changes in the collected current	Does not require immobilization step; relatively fast assay time (30 min); embeddable	Low stability (less than 7 days)	Low	Pre-clinical	[23]
FET with thiol-modified, GA-specific aptamers anchored at gold-coated ZnO nanorods deposited on its gate. Ligand/aptamer complex formation induces changes in the conductivity of the FET visible in the source–drain current vs. source–drain voltage plot	Very fast assay time (few minutes); embeddable	The article lacks a clear identification of a calibration curve, as well as information on important parameters such as the LOD for this technology	Moderate	Laboratory	[101]
PEDOT-coated electrode coupled with a protein-based recognition element allows the measurement of HSA+GA concentration through impedance analysis; a following enzymatic reaction allows quantification of GA alone by measuring current vs. applied voltage	Measures both GA and HSA; relatively fast assay time (15 min); embeddable	Attention to the stability of recognition element and enzyme is crucial	Moderate	Pre-clinical	[24]

**Table 6 biosensors-12-00687-t006:** Additional relevant approaches comparison.

Detection Mechanism	Advantages	Disadvantages	Complexity	Deployment Phase	Ref.
ELISA-like assay based on fluorescent quenching interactions between GO and modified aptamers. In the presence of GA, aptamers detach from GO to bind to target analyte and fluorescent signal previously quenched is restored	Relatively fast assay time (30 min); stability in human serum (DNase resistance)	The readout mechanism can not be easily incorporated in a POC device	Moderate	Pre-clinical	[105]
Paper dipstick strip exposing general albumin aptamers to trap albumin. GA or HSA-specific aptamers/gold nanoparticle complex later attaches to the trapped analyte in a sandwich structure that generates a colorimetric signal	The same dipstick can measure either HSA or GA, it is the aptamers/gold nanoparticle complex solution that discriminates the measurement; long-term stability at room temperature (30 days)	The readout mechanism can not be easily incorporated in a POC device	Low	Laboratory	[106]
Two-strip LIFA with disconnected bridge structure to separate the loading from the sample solution. Target analyte in the sample solution is captured by specific antibodies and then forms a sandwich structure with a-specific antibodies/gold nanoparticle complex transported by the loading solution to generate a colorimetric signal	Measures both GA and HSA; Relatively fast assay time (30 min)	The readout mechanism can not be easily incorporated in a POC device; multiple injections required	Moderate	Laboratory	[107]
Randomly oriented Ag/SiNWs cluster around target molecules and trap them. Raman spectra within these regions provide information on GA concentration in the sample. A chemometric classifier quantifies glycation ratios	Measures both GA and HSA; moderate stability (3 weeks, but signal intensity decreases)	The readout mechanism can not be easily incorporated in a POC device; complex technology (possible high cost)	High	Laboratory	[108]

**Table 7 biosensors-12-00687-t007:** Synthesis of the information and performances of the biosensors for POC glycated albumin analysis presented in this review.

Method	Target	Test Sample	Measurement Range		Technology	Ref.
			Target Absolute Values	Relative GA [%]		
Enzymatic	GA and HSA	20 μL serum	/	9.6–20	Disposable test-strip + portable optical analyzer with temperature control unit and data processing/visualization unit	[97]
	ϵ−FK	0.8 μL Z-FK solution	0.33–3.33 g/dL	/	Disposable IDE + potentiostat (chronoamperometry measurement)	[99]
	GA	0.8 μL reaction mixture	0.13–0.67 g/dL	/	Disposable SPCE + potentiostat (chronoamperometry measurement)	[100]
Affinity	GA or HSA	n.a.	GA: 0.1–42 μg/dL HSA: 0.1–62 μg/dL	/	IDE + potentiostat (EIS measurement)	[20]
	GA or HSA	<1 μL diluted plasma	GA: 0.16 μg/dL–1.6 g/dL HSA: 5 μg/dL-10 g/dL	/	SPCE + potentiostat (SWV measurement)	[21]
	GA	40 μL GA solution	0.2–1 mg/dL	/	Electrochemical cell + potentiostat (SWV measurement)	[22]
	GA	200 μL reaction mixture	1 μg/dL–5 mg/dL	/	SPCE + potentiostat (SWV measurement)	[23]
	GA	n.a.	7.7–33.4 mg/dL	/	FET + voltage control and current acquisition system	[101]
	GA and HSA	n.a.	GA: 33 μg/dL–6.7 mg/dL HSA: 33 μg/dL–6.7 mg/dL	5–80	PEDOT electrode + potentiostat (SWV measurement)	[24]
	GA	Diluted serum	5–30 mg/dL	/	ELASA assay + spectrometric technology (fluorescence measurement)	[105]
	GA or HSA	25 μL reaction mixture in bovine serum	GA: 0–2 g/dL HSA: 0–5 g/dL	/	Paper dipstick assay + optical scanner (colorimetry measurement)	[106]
	GA and HSA	2.5 μL sample solution and 130 μL loading solution	GA: 100 ng/dL–100 mg/dL HSA: 50 μg/dL-360 mg/dL	5–36	LFIA + imaging system (colorimetry measurement)	[107]
	GA and HSA	n.a.	GA: 3.33 mg/dL-0.67 g/dL	5–25	Ag-coated silicon nanowires + laser source and detector (SERS measurement)	[108]

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
