# Peer review of "Analytical Challenges in Diabetes Management: Towards Glycated Albumin Point-of-Care Detection"

_biosensors, 2022, doi:10.3390/bios12090687_

Round 1

Reviewer 1 Report

This review is extensive and well-written.  This reviewer only has minor comments/edits:

Line 31: “…who refer polydipsia…”  The verb should be changed.  One suggestion: “…who experience polydipsia”.

Line 36:  There should be a reference at the end of the sentence (does #1 apply?)

Line 40: “…oral hypoglycemic agents up to insulin??  This wording should be changed.  Do the authors mean “…including insulin”?

Line 98: “…foresees rigorous”  The verb should be changed. 

Line 124: “10-years” -> “10-year”

Line 30: “urine ketones” -> “urine ketone”

Table 2 title: The authors should consider saying “may be more reliable”

Line 212: “at clinical site” -> “at clinical sites”

Line 210: The sentence beginning on this line is quite long and somewhat hard to read

There is a lot of information about the enzymatic and affinity techniques of measuring GA (3.2) and the information presented in Table 4 is quite helpful.  Could the authors consider inserting another column(s) into the table to indicate benefits/limitations and whether any of these techniques are currently being used in any kind of laboratory or clinical setting?  They should also consider referring to the table near the beginning of the written section so the readers know that this information is later summarized.

Line 335: “the authors developed also” -> “the authors also developed”

Line 368: “nucleotides sequences” -> “nucleotide sequences” (same with Table 3 title)

Line 392: “in presence of GA” -> “in the presence of GA”

Line 413: “designe” -> “design”

Line 520: the second conclusion should be reworded.  It is a little difficult to follow. 

Reviewer 2 Report

1.   The author repeats the full name of some abbreviations that have been proposed before in some content, i.e., IGT, IFG.

2.  Although different kinds of Glycated albumin analysis are introduced, their advantages and disadvantages are not compared.

3.    Ref 16, 46, 60, 99, and 116, are missing the page number.

4.  I think the article needs a logical and clear schematic to summarize all the technologies and Glycemic control biomarkers mentioned in the review.

5.  I think more detailed mechanisms of detection are missing in each tech of the review, like lateral flow immunoassay, besides the advantages and main drawbacks to overcome in each approach. There were done in parts of the review, but not covered in all topics.
